# Effect of Multi-Mode Thermosonication on the Microbial Inhibition and Quality Retention of Strawberry Clear Juice during Storage at Varied Temperatures

**DOI:** 10.3390/foods11172593

**Published:** 2022-08-26

**Authors:** Min Feng, Bimal Chitrakar, Jianan Chen, Md. Nahidul Islam, Benxi Wei, Bo Wang, Cunshan Zhou, Haile Ma, Baoguo Xu

**Affiliations:** 1School of Food and Biological Engineering, Jiangsu University, Zhenjiang 212013, China; 2Institute of Food Physical Processing, Jiangsu University, Zhenjiang 212013, China; 3College of Food Science and Technology, Hebei Agricultural University, Baoding 071001, China; 4Department of Agro-Processing, Bangabandhu Sheikh Mujibur Rahman Agricultural University, Gazipur 1706, Bangladesh

**Keywords:** strawberry juice, enzymatic browning, polyphenol oxidase, active ingredients, shelf life

## Abstract

Strawberry juice, which is rich in nutrients and charming flavor, is favored by consumers. To explore whether multi-mode thermosonication (MTS) can ensure the quality stability of strawberry clear juice (SCJ) during storage, the effects of microbial inhibition, enzyme activity, and physicochemical properties of SCJ pretreated by MTS were evaluated during storage at 4, 25, and 37 °C in comparison with thermal pretreatment (TP) at 90 °C for 1 min. The MTS, including dual-frequency energy-gathered ultrasound pretreatment (DEUP) and flat sweep-frequency dispersive ultrasound pretreatment (FSDUP), were conducted at 60 °C for 5 and 15 min, respectively. Results showed that the total phenols, flavonoids, anthocyanins, ascorbic acid, and DPPH free radical scavenging ability of SCJ decreased during the storage period. The control sample of SCJ was able to sage for only 7 days at 4 °C based on the microbiological quality, while the FSDUP and DEUP group extended the storage period up to 21 and 14 days, respectively. The polyphenol oxidase in SCJ pretreated by MTS did not reactivate during the storage period. The MTS remarkably (*p* < 0.05) reduced the color deterioration, browning degree, and nutrient degradation during the storage period. Moreover, the FSDUP group exhibited the maximum shelf life with a minimum loss of quality, demonstrating that it was the most suitable processing method for obtaining high-quality SCJ. It can be concluded that the MTS has the potential to inhibit enzymatic browning, inactivating microorganisms, preserve original quality attributes, and prolong the shelf life of SCJ.

## 1. Introduction

Strawberry (*Fragaria* × *ananassa* Duch.) is a natural functional food rich in bioactive compounds such as phenolic acids, anthocyanins, and flavonoids [1]. It is one of the most economically important, delicious, and widely accepted nutritious fruits due to its sweet-sour taste, juiciness, and nutritiousness [2]. However, the shelf life of fresh strawberries is very short because of their higher moisture content (more than 90% [3]), which limits their industrial development. Strawberry juice is one of such products with a longer shelf life, which not only overcomes the difficulties in the transportation of delicate strawberry fruits but also facilitates the processing of a huge quantity of strawberries during harvesting season. Moreover, with the improvement in consumer demand and health consciousness, carbonated beverages are no longer considered healthy and popular among consumers due to their low nutritional value and high-calorie content [4].

The fruit juices are obtained by mechanical extraction (squeezing) of different fruits. Strawberry juice contains many nutrients such as minerals, vitamins (especially vitamin C), antioxidants, carotenoids, phytochemicals, and dietary fiber, which are essential for human health [5]. The polyphenol oxidase activity from strawberries leads to color and flavor changes, phase separation, viscosity changes, and ultimately spoilage of the juice. A processing method is chosen to inactivate such endogenous enzymes in strawberries so as to control above-mentioned defects; thermal treatment is the most commonly and widely used technique for such purpose. However, the thermal treatment used to inactivate enzymes may cause juice quality degradation, including fade-color and lower flavor, as well as nutritional degradation from the thermal processing [6]. Simultaneously, a novel and efficient processing technology is in demand to fulfill the purpose with minimal degradation of strawberry quality.

Ultrasound is a rapidly emerging non-thermal processing technology, especially for heat-sensitive foods such as strawberry juice, to maintain its original freshness, flavor, and color [7]. However, most of the studies have used ultrasonic cleaners and ultrasonic cell crushers for ultrasonication. These devices generate single-frequency ultrasound waves, which are prone to the formation of standing waves, resulting in an uneven distribution of sound fields with poor cavitation effects. Instead, multi-mode ultrasound realizes alternating combinations of multiple frequencies, pulsed ultrasound generation, and real-time monitoring and control of critical parameters, which can overcome the above-mentioned shortcoming [8,9,10]. The number of cavitation bubbles produced by multi-frequency ultrasonic treatment is five times greater than that of single-frequency ultrasonic treatment [11]. It is a well-known fact that when the ultrasound passes through a medium, the microbubbles create a series of compression and expansion cycles [12]. During transient cavitation, the formed bubbles undergo irregular oscillations, producing regions of temperature and pressure variation, which ultimately leads to cell disruption and enzyme inactivation [13,14].

Other issues with ultrasound treatment are that it cannot always achieve complete enzyme inactivation and/or microbial destruction during normal temperature treatment. For example, ultrasound treatment retained the overall quality of strawberry juice but was unable to reduce the polyphenol oxidase (PPO) activity [15]. In another study with mango juice, ultrasonic treatment destructed the coliforms to an undetectable level; however, bacteria, yeast, and mold remained active even after such treatment for up to 60 min [16]. Based on these results, the thermosonication concept was proposed, where ultrasound treatment was applied at a slightly higher temperature of more than 50 °C [17]. Such treatment combines ultrasonic cavitation effect with thermal effect, which can effectively inactivate endogenous enzymes and microorganisms at lower temperature and shorter treatment time without affecting the quality of the product [18,19]. Dundar et al. [20] combined ultrasound and thermal treatment in strawberry juice, and the residual enzyme activity of PPO was reduced from 63.8% to 11.4% at the temperature range of 25 to 75 °C.

Our recent study on ultrasound treatment of strawberry juice (at 50 °C and 30 min) demonstrated the reduction in PPO activity by 21.8%, 10.2%, and 1.5% at frequencies of 22, 40, and 22 + 40 kHz, respectively [2]. In order to investigate the effects of MTS on the overall quality of strawberry juice and its stability during storage, this study proposed the ultrasound treatment in two different modes, namely flat sweep-frequency dispersive ultrasound pretreatment (FSDUP) and dual-frequency energy-gathered ultrasound pretreatment (DEUP). Such treated juice was studied for its storage stability at three different temperatures of 4, 25, and 37 °C, considering microbial quality, enzyme activity, and physicochemical properties as the indicators.

## 2. Materials and Methods

### 2.1. Raw Materials and Chemical Reagents

The strawberries (*Fragaria × ananassa* Duch. cv. HongYan) were purchased from a strawberry farm in Danyang, Jiangsu, China. Mature strawberries of uniform size and color without mechanical damage/fingerprints were selected as the experimental subjects and kept under cold storage (4 °C) until they were used. All reagents such as methanol, gallic acid, Folin–Ciocalteu, rutin, and vitamin C were obtained from Sinopharm Chemical Reagent Co., Ltd. (Shanghai, China).

### 2.2. Preparation and Processing of Strawberry Clear Juice

The fresh strawberries were washed carefully to remove any adhering dirt or impurities. The calyx and stems were also removed. Then, they were manually cut into small cubes with a sterile knife, added water at a liquid-to-material ratio of 2:1 mL/g, followed by juicing using a beater (HB500A, Hauswirt, Qingdao, China). Then, the enzymatic hydrolysis of strawberry juice was performed based on the pre-experimentally optimized enzymatic process (enzyme addition of 0.17%; pectinase to cellulase ratio of 3:1; enzymatic temperature of 41 °C; and enzymatic reaction time of 35 min). After enzymatic hydrolysis and centrifuged at 8000× *g* for 10 min, the supernatant was filtered through two layers of gauze to obtain SCJ.

Based on our recent work on the enzyme inactivation parameters [2], thermal pretreatment (TP) and multi-mode thermosonication (MTS) were performed on SCJ. As shown in Figure 1A, flat sweep-frequency dispersive ultrasound pretreatment (FSDUP) was performed at 60 °C for 15 min under the frequency of 20 + 40 kHz in simultaneous operation mode, while dual-frequency energy-gathered ultrasound pretreatment (DEUP) was performed at 60 °C for 5 min under the frequency of 20/40 kHz in sequential operation mode (Figure 1B). For thermal pretreatment (TP), the SCJ was heated in a water bath at 90 °C for 1 min, while the control group was the SCJ without any treatment.

### 2.3. Determination of Microorganisms

The total number of colonies as well as molds and yeasts were used as microbial indexes during storage in this study. The total number of colonies was determined using the plate count method (using agar medium and incubated at 36 °C for 48 h). Molds and yeasts were incubated in rose bengal medium at 28 °C for 3–5 days. The results were expressed as the logarithms of the average number of colony-forming units per mL (log CFU/mL) [21].

### 2.4. Determination of pH and Total Soluble Solids (TSS)

The pH of SCJ was measured using an HI 991,001 pH m (Hanna Instruments, Woonsocket, RI, USA). The TSS content was determined using a digital refractometer (PAL-3, Atago, Tokyo, Japan) and expressed as °Brix.

### 2.5. Determination of Polyphenol Oxidase (PPO) Activity

Preparation of crude enzyme solution: 50 mL of strawberry pulp were added to an equal volume of phosphate buffer (pH 6.9) containing 2% polyvinylpyrrolidone and then placed at 4 °C for 1 h. The supernatant obtained after centrifugation for 15 min at 11,000× *g* and 4 °C was used for the determination of PPO activity. In 0.1 mL of such extracted crude enzyme, 2.9 mL of catechol solution (0.05 mol L^−1^) was mixed, and the absorbance was measured at 420 nm for 3 min [22]. The unit of enzyme activity was defined as the amount of enzyme that causes a change of 0.001 absorbance units per minute, and PPO activity was expressed as ∆OD/(mL·min). The results were expressed as residual enzyme activity, and the residual activity of PPO was calculated as:(1)$$Residual\;PPO\;activity\;\left(\%\right)=\frac{A_{t}}{A_{0}}\times 100\%$$
where *A_t_* represents the remaining enzymatic activity at time *t*, and *A*_0_ is the initial PPO activity of strawberry juice before treatment.

### 2.6. Analysis of Color Properties

#### 2.6.1. Determination of Color Difference

The color of 30 mL SCJ was determined by a colorimeter (CR-400, Konica Minolta, Inc., Tokyo, Japan). Then the color values, namely *L** (brightness), *a** (red/green), and *b** (yellow/blue) of control and treated SCJ were measured at room temperature, and the total color difference (Δ*E*) was calculated by using the following equation:(2)$$\Delta E=\sqrt{{\left(L^{*}-L_{0}^{*}\right)}^{2}+{\left(a^{*}-a_{0}^{*}\right)}^{2}+{\left(b^{*}-b_{0}^{*}\right)}^{2}}$$
where *L*_0_***, *a*_0_***, and *b*_0_*** are, respectively, the brightness, red/green, and yellow/blue values of fresh strawberry juice, while *L**, *a**, and *b** are those of treated SCJ.

#### 2.6.2. Determination of Browning

According to the method of Wu [23], 5 mL of SCJ was mixed with an equal amount of distilled water and then centrifuged at 11,000× *g* for 10 min, and the supernatant was taken to determine its absorbance at 420 nm by ultraviolet spectrophotometer (UV-1801, Beijing PUXI General Instrument Co., Ltd., Beijing, China).

#### 2.6.3. Determination of Clarity

The clarity of SCJ was expressed as the transmittance (T, %) by measuring the transmittance of the juice at 660 nm using an ultraviolet spectrophotometer (UV-1801, Beijing PUXI General Instrument Co., Ltd., Beijing, China) [24].

### 2.7. Determination of Active Ingredients

#### 2.7.1. Determination of Total Phenolic Content (TPC)

The TPC was determined according to the method described by Xu et al. [11] with some modifications. Briefly, 0.5 mL of diluted SCJ (50 times dilution) was added to 2.5 mL of 5% Folin–Ciocalteu reagent and left for 2 min. Then, 2 mL of 7.5% Na_2_CO_3_ solution was added, and the reaction was carried out at 50 °C for 5 min. The absorbance value was measured at 760 nm. A standard curve of total phenols was drawn with gallic acid as the standard, and the results were expressed as gallic acid equivalent (mg GAE/100 mL).

#### 2.7.2. Determination of Total Flavonoid Content (TFC)

According to the method of Xu et al. [25], 2 mL of diluted SCJ (5 times dilution) was mixed with 0.3 mL of 5% NaNO_2_ solution. After 6 min of reaction in a dark place, 0.3 mL of 10% Al(NO_3_)_3_ was added and left for 6 min. Then, 2 mL of 4% NaOH was added; then added 4.9 mL of distilled water. After standing for 10 min, the absorbance was measured at 510 nm with a spectrophotometer (T6NC, Beijing PUXI General Instruments Co., Ltd., Beijing, China). The results were expressed as rutin equivalents (mg RE/100 mL) from the standard curve drawn with rutin standard.

#### 2.7.3. Determination of Total Anthocyanin Content (TAC)

The TAC was determined by the pH differential method according to Xu et al. [1]. Firstly, 2 mL of SCJ was mixed with 8 mL of pH 1.0 buffer solution (0.1 M KCl) and pH 4.5 buffer solution (0.5 M NaAc). Then, the absorbance values were determined at 510 nm and 700 nm using a spectrophotometer (T6NC, Beijing PUXI General Instruments Co., Ltd., Beijing, China). The results were expressed as cyanidin-3-glucoside equivalents (mg CGE/100 mL).

#### 2.7.4. Determination of Ascorbic Acid Content (AAC)

The AAC was determined according to Zuo et al. [26] using the HPLC chromatography technique with some modifications. Briefly, SCJ was mixed with 0.1% oxalic acid solution at a ratio of 1:5 and extracted at 4 °C for 2 h. The sample was filtered through a 0.45 μm membrane and then detected by high-performance liquid chromatography (HPLC) running with a C18 column (4.6 mm × 250 mm, 5 μm). The HPLC operational parameters were as follows: column temperature of 25 °C, the mobile phase of 0.1% oxalic acid isocratic solution, the flow rate of 0.8 mL/min, the detection wavelength of 254 nm, and the injection volume of 20 μL. Ascorbic acid of SCJ was identified by comparisons of the retention time (t_R_ = 7.799 min) with the ascorbic acid standard, and the concentration was quantified by the external calibration method. The calibration curve of ascorbic acid (y = 85,838x – 60,180, R_2_ = 0.9998) was constructed by plotting the peak areas versus the concentrations (from 0 to 0.25 g/L) of the standard compound. The results were expressed as mg ascorbic acid per 100 mL of juice.

### 2.8. Determination of Antioxidant Activity

#### 2.8.1. Analysis of DPPH Free Radical Scavenging Capacity

The DPPH free radical scavenging capacity was measured using the method of Xu et al. [11]. A total of 0.4 mL of SCJ was mixed with 3.6 mL of 0.14 mM DPPH solution, and then the reaction was carried out for 30 min before the absorbance was measured at 517 nm.

#### 2.8.2. Analysis of ABTS Free Radical Scavenging Capacity

Following the method described by Cortés-Macías et al. [27], the ABTS free radical scavenging capacity was measured. A total of 0.4 mL of SCJ was mixed thoroughly with 3.6 mL of ABTS solution. After reacting in the dark for 10 min, the absorbance was measured at 734 nm.

### 2.9. Statistical Analysis

All measurements were carried out in triplicate, and the results were expressed as mean ± standard deviation. One-way analysis of variance (ANOVA) was performed using SPSS 20.0 software (IBM, Chicago, IL, USA). The significant difference between the means was determined by using Duncan’s test procedure at a 95% confidence level (*p* < 0.05).

## 3. Results and Discussion

### 3.1. Analysis of Microorganisms of Strawberry Clear Juice during Storage

The microbial load can reflect the freshness of a juice, which is often used as a basic indicator of its shelf life. Table 1 shows the microbial counts (total number of colonies and mold and yeast counts) in SCJ during storage at 4, 25, and 37 °C. The results showed that both the microbial counts were found at undetected levels (ND) at various storage periods; the length of such period was dependent on the type of treatment applied and the storage temperature. For example, the storage temperature of 4 °C resulted in the longest ND period as against the shortest ND for 37 °C, while in terms of the type of treatment, the order of ND period was TP > FSDUP > DEUP. More specifically, the storage of CSJ at 4 °C for 28 days showed total colony counts of 1.25, 1.89, 2.72, and 3.10 log CFU/mL for TP, FSDUP, DEUP, and control, respectively, while the storage at 37 °C for the same period showed 2.01, 2.24, 4.95 and 5.85 log CFU/mL, respectively. These results indicated that lower temperature storage can inhibit microbial reproduction. In combination with the lower temperature, the FSDUP and DEUP groups stored at 4 °C were able to extend the shelf life up to 21 and 14 days, respectively. Fan et al. [28] also confirmed that carrot juice processed by thermosonication at 52 °C showed significant microbial growth stability during storage at 6 °C and consequently extended the product shelf life. The ultrasonic cavitation might have disrupted the integrity of cell membranes and damaged the nucleic acids, leading to cell lysis and death [29,30]. However, the increase in microbial counts with the extension of storage time might be due to the recovery of the damaged cells and the growth of the surviving cells [21]. In summary, it was found that the MTS combined with lower temperature storage improved the microbial stability of CSJ.

### 3.2. Analysis of pH of Strawberry Clear Juice during Storage

The pH of juice can reflect the degree of deterioration of juice quality. The pH changes of SCJ under different treatment conditions during storage are shown in Figure 2A. The pH of SCJ was relatively stable during storage at 4 °C. When stored at 37 °C, the pH of all samples showed a decreasing trend, especially in the control group. This was mainly due to the decomposition of carbohydrates by some residual acid-producing microorganisms to produce various organic acids, leading to a drop in pH [31]. Adedokun et al. [32] also reported that the pH values of fruit juices decreased during storage due to the spoilage microorganisms.

### 3.3. Analysis of Total Soluble Solids Content of Strawberry Clear Juice during Storage

The TSS content is the general term for all water-soluble compounds in food. The changes of TSS in SCJ under different treatments during storage are shown in Figure 2B. The results showed that the TSS content of SCJ increased first and then decreased during storage, especially in the TP group, which might be related to the continuous hydrolysis of reducing sugars under acidic conditions [33]. A similar finding was reported by Raji et al. [34] for thermally processed pineapple and bitter orange mixed fruits juices stored at room temperature. The TSS content was negatively correlated with storage temperature, and the TSS of strawberry juice at 4 °C was relatively stable, while it showed a decreasing trend with increasing storage temperature (at 25 and 37 °C). This was most probably due to the rapid multiplication of microorganisms in the juice at higher temperatures with increasing storage time. Moreover, the TSS is used as a nutritional source for microbial fermentation, leading to its significant decrease [21].

### 3.4. Analysis of PPO Activity of Strawberry Clear Juice during Storage

Our previous study found that MTS significantly (*p* < 0.05) reduced PPO activity but did not completely inactivate it, indicating its possibility to reactivate during storage [2]. Therefore, it is necessary to observe the changes in PPO activity in SCJ during storage. It can be seen from Table 2 that the PPO activity of TP and MTS groups showed some fluctuations during storage; however, the changes were not obvious. No reactivation of PPO was observed in TP and MTS groups, which might be due to the irreversible damage to PPO caused by TP and MTS. Additionally, the changes in pH during storage further inhibited the enzyme activity. Similar results were observed by Suo et al. [7], where the enzyme activity of TP and MTS pumpkin juice was low and stable during storage. However, the residual enzyme activities of the control group at the end of storage were 48.47% (4 °C), 40.27% (25 °C), and 28.28% (37 °C). These residual PPO activities were closely associated with color changes, leading to the degradation of bioactive components [2]. This, in turn, led to a reduction in the sensory quality and nutritional properties of SCJ.

### 3.5. Analysis of Color Properties of Strawberry Clear Juice during Storage

#### 3.5.1. Analysis of the Color Difference

Color is an essential sensory criterion for consumers to measure the overall acceptability of fruit juices and an important indicator of their freshness as well as their hygienic condition [35]. The color depends on the type and content of natural pigments in the juice, while enzyme activity and microbial content also affect it [31]. The color changes of SCJ under different storage conditions are shown in Table 3. The color of SCJ was mainly composed of natural pigments such as anthocyanin, and its degradation and isomerization of anthocyanin with the extension of storage time led to a gradual decrease in *a** value. Buvé et al. [36] also reported that the *a** value of pasteurized strawberry juices decreased continuously during storage. In the control group, the ∆*E* of SCJ stored at 4, 25, and 37 °C for 28 days was 6.53, 24.07, and 25.29, respectively. This indicated that the storage temperature was an important factor in the color change of SCJ. The changes in ∆*E* might be due to the acceleration of the Maillard reaction with the increase in temperature, resulting in severe browning [37]. Comparing the different treatments, MTS better preserved the color of SCJ during storage. This change is also evident in Figure 3. Especially, the flat sweep-frequency dispersive ultrasound, with uniform sound field distribution, has the least effect on the color of SCJ. These changes may be closely related to chemical, biochemical, enzymatic, and physical changes during ultrasonic processing [15,16,17]. Therefore, FSDUP combined with low-temperature storage has a suitable effect on maintaining the color of SCJ.

#### 3.5.2. Analysis of Browning and Clarity

The changes in browning of SCJ under different storage conditions are shown in Table 4. The browning of SCJ in the control group was 0.17, and the browning after TP, FSDUP, and DEPU was 0.26, 0.22, and 0.23, respectively. The increase in browning was related to the increase in substances released from the cells by the TP and MTS [38]. The browning of SCJ increased significantly during storage, which was consistent with the changes in ∆*E* values (Table 1). It is also evident from Figure 3 that the SCJ gradually changed from red to pink during storage. With the increase in storage temperature, it eventually turned yellowish brown, indicating that significant browning occurred during storage. The increase in browning was mainly due to non-enzymatic browning, which is the occurrence of the Maillard reaction between reducing sugars and amino acids [39]. The same phenomenon was also observed in pumpkin juice [7] and cucumber juice [40].

As shown in Table 4, the transmittance of the control group decreased by 21.17%, 42.23%, and 49.90% during storage at 4, 25, and 37 °C, respectively. The higher the storage temperature, the faster the decrease in the clarity; this phenomenon might be due to the higher temperature leading to accelerated microbial growth, making SCJ turbid. However, MTS was able to effectively kill microorganisms, which resulted in a minimum decrease in the clarity of SCJ during the storage period. In particular, the transmittance of SCJ treated by swept-frequency ultrasound was higher, indicating that the juice was more clarified.

### 3.6. Analysis of Active Ingredients of Strawberry Clear Juice during Storage

#### 3.6.1. Analysis of Changes in Total Phenolic Content (TPC)

The changes in the TPC of SCJ during storage are shown in Figure 4A. The TPC of the control group decreased by 46.10%, while that of the TP, FSDUP, and DEUP groups decreased by 66.01%, 47.22%, and 55.59%, respectively, during storage at 4 °C for 28 days. The higher retention of TPC in FSDUP and DEUP groups, as compared to the TP group, might be attributed to the cavitation effect of ultrasound, which eliminated dissolved oxygen and delayed the oxidative degradation of polyphenols during storage [35,41]. Comparing the changes in TPC of SCJ at three storage temperatures, it was found to decrease slowly at 4 °C; the decline rate was accelerated at 25 °C in the later stage of storage. At 37 °C, the TPC sharply reduced at the early stage and then tended to be gentle. Nowicka et al. [42] also found that the oxidative degradation of polyphenols was evident at 25 and 37 °C, which was relatively insignificant at 4 °C. Therefore, storage at 4 °C was found to be suitable for maintaining the TPC of strawberry juice, and FSDUP better preserved the quality of strawberry juice.

#### 3.6.2. Analysis of Changes in Total Flavonoid Content (TFC)

The changes in the TFC of TP and MTS groups during storage are shown in Figure 4B. After storage at 4 °C for 28 days, the TFC of the control group decreased by 49.47%, while that of the TP, FSDUP, and DEUP groups decreased by 58.33%, 52.86%, and 55.42%, respectively. In other words, TP led to a drastic loss of TFC, while MTS effectively preserved flavonoids. Especially, FSDUP resulted in the least loss of TFC in SCJ during storage. Therefore, the ultrasonic cavitation effect improved the stability of flavonoids in SCJ during storage. These observations are in agreement with the findings of Fan et al. [28], where the ultrasound-treated carrot juice better maintained the nutritional properties compared to the control group. A longitudinal comparison of the three storage temperatures showed that the higher the temperature, the faster the degradation of flavonoids; therefore, low-temperature storage is more suitable for the retention of flavonoids in SCJ.

#### 3.6.3. Analysis of Changes in Total Anthocyanin Content (TAC)

The anthocyanin, which determines the color of SCJ, is extremely unstable and susceptible to the effects of pH, temperature, oxygen, and metal ions. The anthocyanin contents of the treated CSJ, stored at different temperatures, are shown in Figure 5A. The anthocyanin content of the control group was 9.18 mg/100 mL, while that of the TP, FSDUP, and DEUP groups was 7.47, 8.82, and 8.65 mg/100 mL, respectively. The TAC of all treated groups decreased significantly (*p* < 0.05) during storage; the decrease was slow in the early storage period, which was accelerated in the later storage period. After 28 days of storage at 4 °C, the anthocyanin retention rate of the FSDUP group was the highest (74.61%), followed by the DEUP group (70.88%). The results were significantly higher than that of TP (69.39%) and the control group (62.40%). The retention of anthocyanins in MTS groups was higher than that in the TP group, indicating that MTS was beneficial to improving and maintaining the stability of anthocyanins in SCJ. Wong et al. [43] also reported that anthocyanins could remain relatively stable under MTS.

#### 3.6.4. Analysis of Changes in Ascorbic Acid Content

The changes in the ascorbic acid content of SCJ during storage at 4, 25, and 37 °C after different treatments are shown in Figure 5B. Ascorbic acid is an important nutrient in fruits that is highly sensitive to heat and oxygen, making it easily degraded [44]. As can be seen from Figure 5B, the degradation of ascorbic acid during storage was very significant (*p* < 0.05) compared to other active ingredients. For example, after 28 days of storage at 4 °C, the retention of ascorbic acid content in the control group was 6.20%, while that in the TP, FSDUP, and DEUP groups was 16.64%, 54.13%, and 42.37%, respectively. The degradation of ascorbic acid during storage was mainly due to oxidative degradation, where the oxygen remaining in the strawberry juice at the beginning of storage led to aerobic degradation, followed by an anaerobic degradation when the oxygen was completely depleted [3]. MTS had the effects of dissolved oxygen and enzyme inactivation, which could effectively reduce the degradation of ascorbic acid during storage [17]. Tiwari et al. [45] also found higher ascorbic acid retention (78.60%) for sonicated strawberry juices at the highest acoustic energy density level (0.81 W/mL) and treatment time (10 min) during 10 days of storage at 4 °C compared to untreated samples at the same storage conditions (76.20%). Additionally, compared with the juice stored at 25 and 37 °C, the ascorbic acid content in the juice stored at 4 °C showed higher stability. In summary, MTS combined with low-temperature storage can significantly (*p* < 0.05) improve the retention of bioactive substances.

### 3.7. Analysis of the Antioxidant Activity of Strawberry Clear Juice during Storage

The positive impact of juices on human health is mainly due to their high antioxidant capacity, which can protect the cells from oxidative damage. The changes in the DPPH and ABTS radical scavenging capacity of SCJ during storage are shown in Figure 6. During storage, the antioxidant activity of SCJ showed a decreasing trend, which was consistent with the trend of TPC (Figure 4). A significant (*p* < 0.05) correlation between TPC and antioxidant properties was observed in orange juice [46]. At different storage temperatures, the antioxidant capacity of the MTS groups was consistently higher than that of the TP group. This is because the ultrasonication generated free hydroxyl radicals and produced hydroxylated derivatives, which improved the antioxidant activity [35]. The DPPH radical scavenging capacity was better retained in SCJ during storage at 4 °C, while it was severely lost at 25 and 37 °C. Similarly, the ABTS free radical scavenging capacity showed the same trend; its value for the control group decreased by 19.84%, 27.14%, and 33.59% after 28 days of storage at 4, 25, and 37 °C, respectively. Therefore, the higher the storage temperature, the higher the loss of antioxidant capacity of SCJ, which was probably due to the accelerated rate of oxidative degradation of active ingredients caused by the higher temperature.

## 4. Conclusions

In conclusion, this study investigated the effects of MTS and storage conditions on the storage stability and quality properties of SCJ. The results showed that the total number of colonies as well as mold and yeast counts in the control group increased significantly (*p* < 0.05) during the storage period. The MTS was able to prolong the storage period; more specifically, the FSDUP group met the juice safety standard at 4 °C storage for 21 days. During storage, pH and TSS were relatively stable, and no reactivation of PPO was detected. The MTS reduced the influence on color change, degree of browning, and juice clarity by passivating PPO, as well as significantly (*p* < 0.05) reduced the enzymatic browning of SCJ. In addition, the active ingredients and antioxidant activity of SCJ were degraded continuously with the extension of storage time, but the active ingredients and antioxidant capacity of MTS groups were significantly (*p* < 0.05) higher than those of TP and the control groups under storage at 25 and 37 °C. The quality of SCJ was less damaged by FSDUP compared with DEUP. Storage at 4 °C was more favorable to maintain the quality properties of SCJ. Therefore, it can be concluded that MTS prolongs the shelf life of SCJ and has little effect on quality properties. FSDUP combined with 4 °C storage is a suitable processing and storage method for SCJ. However, the changes in volatile components during storage need to be further studied to determine the accurate shelf life of SCJ.

## Figures and Tables

**Figure 1 foods-11-02593-f001:**
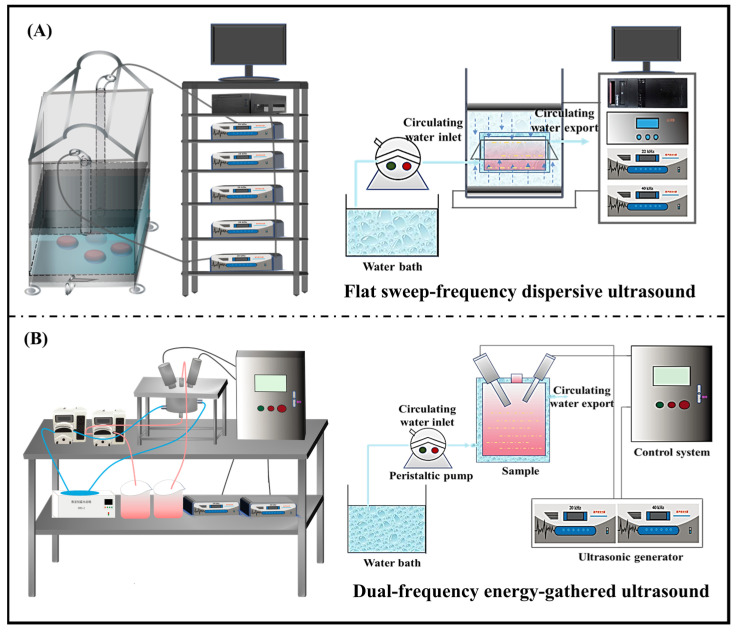
Design of multi-frequency ultrasound devices. (**A**) Flat sweep-frequency divergent ultrasound. (**B**) Dual-frequency energy-gathered ultrasound.

**Figure 2 foods-11-02593-f002:**
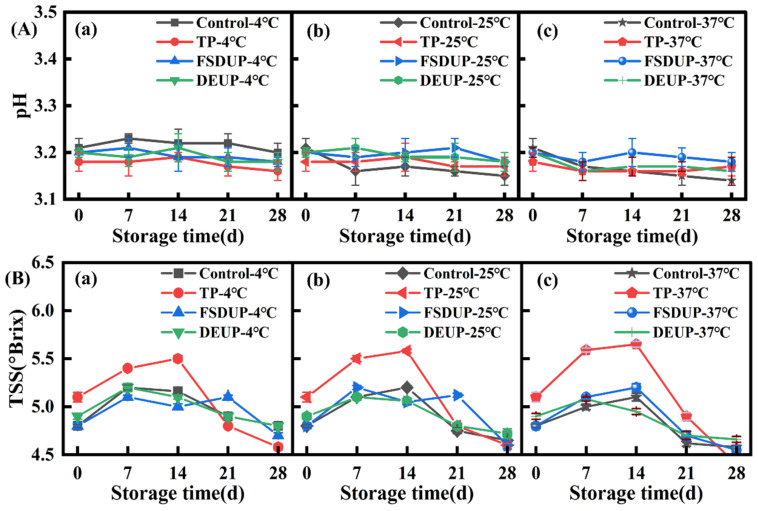
Changes in pH (**A**) and total soluble solids (**B**) of strawberry clear juice during storage after thermal pretreatment and multi-mode thermosonication. TSS: total soluble solids; TP: thermal pretreatment; FSDUP: flat sweep-frequency dispersive ultrasound pretreatment; DEUP: dual-frequency energy-gathered ultrasound pretreatment.

**Figure 3 foods-11-02593-f003:**
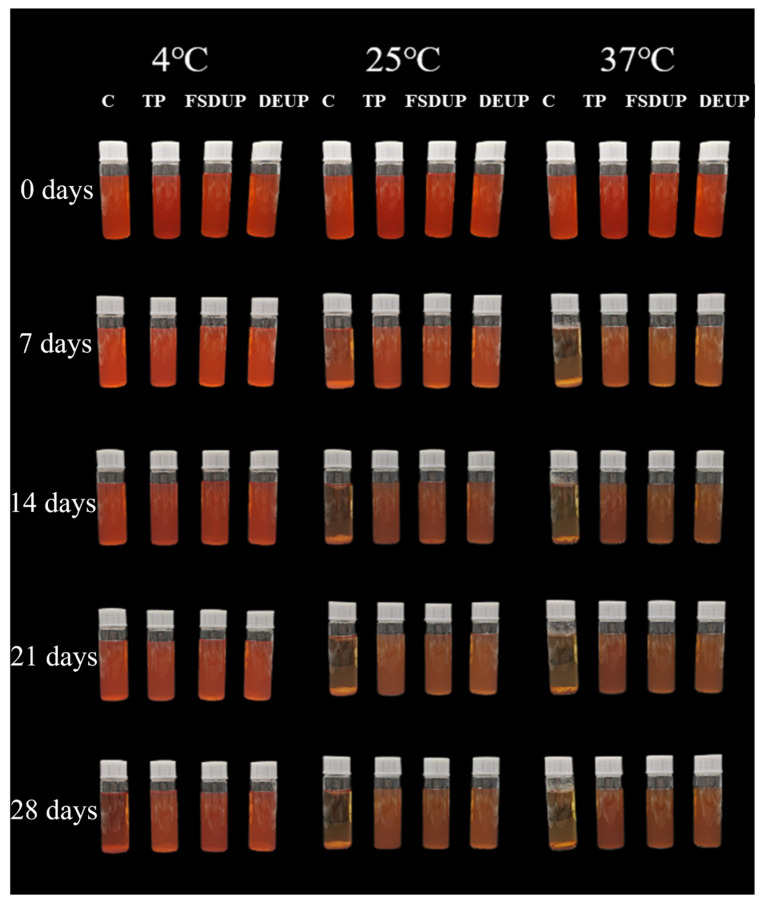
Changes in apparent morphology of strawberry clear juice during storage after thermal pretreatment and multi-mode thermosonication. C: control; TP: thermal pretreatment; FSDUP: flat sweep-frequency dispersive ultrasound pretreatment; DEUP: dual-frequency energy-gathered ultrasound pretreatment.

**Figure 4 foods-11-02593-f004:**
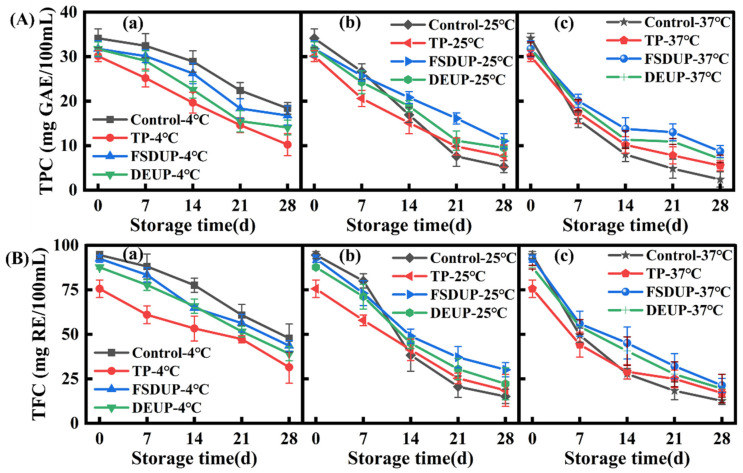
Changes in total phenolic content (**A**) and total flavonoid content (**B**) of strawberry clear juice during storage after thermal pretreatment and multi-mode thermosonication. TPC: total phenolic content; TFC: total flavonoid content; TP: thermal pretreatment; FSDUP: flat sweep-frequency dispersive ultrasound pretreatment; DEUP: dual-frequency energy-gathered ultrasound pretreatment.

**Figure 5 foods-11-02593-f005:**
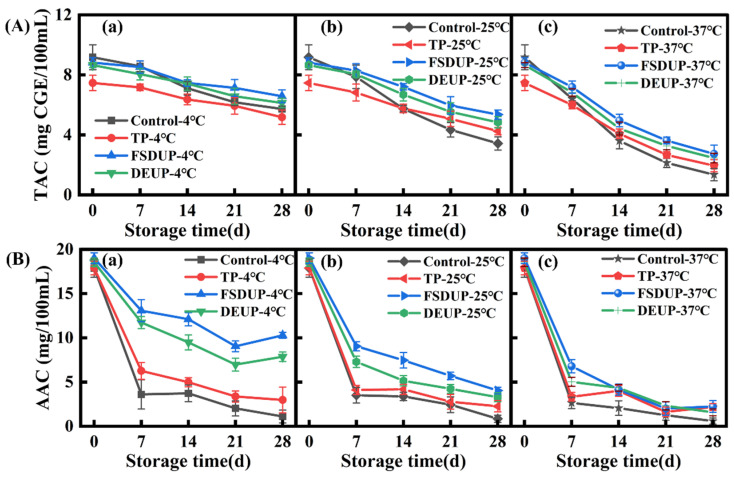
Changes in total anthocyanin content (**A**) and ascorbic acid content (**B**) of strawberry clear juice during storage after thermal pretreatment and multi-mode thermosonication. TAC: total anthocyanin content; AAC: ascorbic acid content; TP: thermal pretreatment; FSDUP: flat sweep-frequency dispersive ultrasound pretreatment; DEUP: dual-frequency energy-gathered ultrasound pretreatment.

**Figure 6 foods-11-02593-f006:**
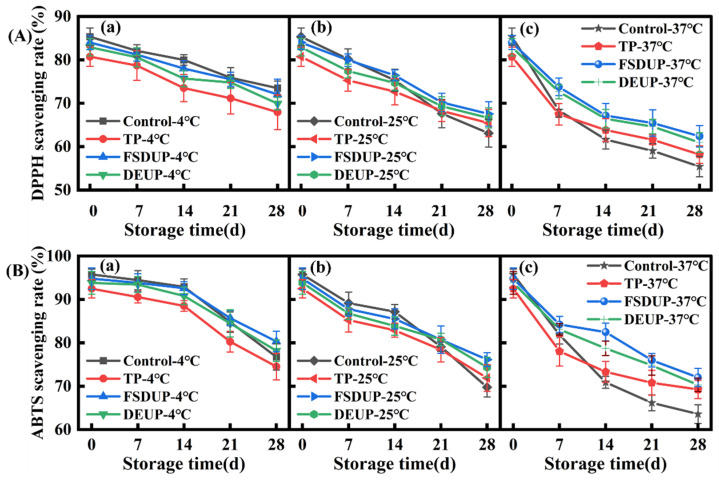
Changes in DPPH (**A**) and ABTS (**B**) free radical scavenging capacity of strawberry clear juice during storage after thermal pretreatment and multi-mode thermosonication. TP: thermal pretreatment; FSDUP: flat sweep-frequency dispersive ultrasound pretreatment; DEUP: dual-frequency energy-gathered ultrasound pretreatment.

**Table 1 foods-11-02593-t001:** Changes in microbial population in strawberry clear juice during storage after thermal pretreatment and multi-mode thermosonication.

Temperature	Time (Days)	Total Number of Colonies (log CFU/mL)				Mold and Yeast Counts (log CFU/mL)			
		Control	TP	FSDUP	DEUP	Control	TP	FSDUP	DEUP
4 °C	0	1.49 ± 0.09 ^e^	ND	ND	ND	1.02 ± 0.05 ^de^	ND	ND	ND
	7	1.58 ± 0.06 ^e^	ND	ND	<1 ^e^	1.25 ± 0.13 ^d^	ND	ND	<1 ^e^
	14	2.02 ± 0.08 ^de^	ND	<1 ^c^	1.14 ± 0.15 ^d^	1.66 ± 0.15 ^cd^	ND	ND	<1 ^e^
	21	2.67 ± 0.06 ^d^	<1 ^c^	1.61 ± 0.03 ^ab^	2.11 ± 0.05 ^cd^	1.87 ± 0.05 ^cd^	ND	<1 ^d^	1.17 ± 0.11 ^d^
	28	3.10 ± 0.13 ^cd^	1.25 ± 0.19 ^b^	1.89 ± 0.14 ^ab^	2.72 ± 0.14 ^c^	2.29 ± 0.07 ^c^	1.10 ± 0.14 ^b^	1.59 ± 0.09 ^bc^	2.12 ± 0.15 ^c^
25 °C	0	1.49 ± 0.09 ^e^	ND	ND	ND	1.02 ± 0.05 ^de^	ND	ND	ND
	7	1.91 ± 0.03 ^de^	ND	ND	1.32 ± 0.13 ^d^	2.11 ± 0.10 ^c^	ND	ND	1.42 ± 0.18 ^cd^
	14	2.88 ± 0.08 ^d^	<1 ^c^	<1 ^c^	2.25 ± 0.05 c^d^	2.95 ± 0.06 ^bc^	ND	<1 ^d^	2.40 ± 0.13 ^c^
	21	3.91 ± 0.12 ^c^	1.34 ± 0.05 ^b^	1.78 ± 0.03 ^ab^	3.04 ± 0.08 ^c^	4.23 ± 0.10 ^ab^	1.32 ± 0.09 ^b^	1.69 ± 0.07 ^bc^	3.57 ± 0.05 ^b^
	28	5.05 ± 0.03 ^b^	1.86 ± 0.04 ^ab^	2.12 ± 0.04 ^a^	3.91 ± 0.07 ^b^	5.01 ± 0.12 ^a^	1.90 ± 0.09 ^a^	2.48 ± 0.09 ^a^	4.43 ± 0.09 ^a^
37 °C	0	1.49 ± 0.09 ^e^	ND	ND	ND	1.02 ± 0.05 ^de^	ND	ND	ND
	7	2.63 ± 0.03 ^d^	ND	ND	1.93 ± 0.13 ^cd^	1.52 ± 0.10 ^cd^	ND	ND	<1 ^e^
	14	3.87 ± 0.05 ^c^	1.14 ± 0.09 ^b^	1.15 ± 0.11 ^b^	2.94 ± 0.05 ^c^	2.23 ± 0.03 ^c^	ND	<1 ^d^	1.79 ± 0.13 ^cd^
	21	4.95 ± 0.05 ^b^	1.57 ± 0.15 ^ab^	1.83 ± 0.08 ^ab^	3.77 ± 0.06 ^b^	3.06 ± 0.06 ^bc^	<1 ^c^	1.38 ± 0.14 ^c^	2.21 ± 0.17 ^c^
	28	5.85 ± 0.06 ^a^	2.01 ± 0.04 ^a^	2.24 ± 0.08 ^a^	4.95 ± 0.15 ^a^	3.96 ± 0.08 ^b^	1.71 ± 0.03 ^ab^	1.97 ± 0.07 ^b^	3.92 ± 0.10 ^ab^

Note: The different lowercase superscript letters (a–e) in the same column indicate significant differences according to Duncan’s test (*p* < 0.05). ND indicates not detected. TP: thermal pretreatment; FSDUP: flat sweep-frequency dispersive ultrasound pretreatment; DEUP: dual-frequency energy-gathered ultrasound pretreatment.

**Table 2 foods-11-02593-t002:** Changes in PPO activity in strawberry clear juice during storage after thermal pretreatment and multi-mode thermosonication.

Temperature	Time (Days)	Relative Residual PPO Activity (%)			
		Control	TP	FSDUP	DEUP
4 °C	0	71.26 ± 2.45 ^a^	2.21 ± 0.14 ^a^	7.11 ± 0.28 ^a^	6.62 ± 0.44 ^a^
	7	61.51 ± 2.75 ^b^	1.53 ± 0.05 ^c^	5.13 ± 0.15 ^bc^	5.78 ± 0.32 ^bc^
	14	62.43 ± 3.83 ^b^	1.88 ± 0.10 ^b^	5.36 ± 0.35 ^bc^	6.36 ± 0.44 ^ab^
	21	54.76 ± 3.54 ^c^	1.44 ± 0.04 ^c^	5.22 ± 0.23 ^bc^	5.84 ± 0.21 ^bc^
	28	48.47 ± 1.31 ^de^	0.56 ± 0.07 ^e^	4.43 ± 0.53 ^de^	4.88 ± 0.10 ^d^
25 °C	0	71.26 ± 2.45 ^a^	2.21 ± 0.14 ^a^	7.11 ± 0.28 ^a^	6.62 ± 0.44 ^a^
	7	54.91 ± 2.93 ^c^	2.37 ± 0.26 ^a^	5.50 ± 0.34 ^b^	5.40 ± 0.23 ^cd^
	14	52.18 ± 1.50 ^cd^	1.43 ± 0.28 ^c^	4.34 ± 0.45 ^de^	5.35 ± 0.44 ^cd^
	21	44.31 ± 2.58 ^ef^	1.07 ± 0.11 ^d^	4.10 ± 0.08 ^ef^	4.86 ± 0.14 ^d^
	28	40.27 ± 2.8 ^fg^	1.64 ± 0.11 ^bc^	4.79 ± 0.16 ^cd^	4.71 ± 0.26 ^de^
37 °C	0	71.26 ± 2.44 ^a^	2.21 ± 0.14 ^a^	7.11 ± 0.28 ^a^	6.62 ± 0.44 ^a^
	7	44.77 ± 3.09 ^ef^	1.58 ± 0.09 ^c^	5.10 ± 0.09 ^bc^	6.61 ± 0.43 ^a^
	14	45.48 ± 3.36 ^ef^	1.13 ± 0.11 ^d^	4.21 ± 0.15 ^de^	4.10 ± 0.14 ^ef^
	21	34.77 ± 3.28 ^g^	0.93 ± 0.05 ^d^	3.86 ± 0.12 ^ef^	4.14 ± 0.10 ^ef^
	28	28.28 ± 2.26 ^h^	0.50 ± 0.08 ^e^	3.52 ± 0.35 ^f^	3.77 ± 0.32 ^f^

Note: The different lowercase superscript letters (a–h) in the same column indicate significant differences according to Duncan’s test (*p* < 0.05). TP: thermal pretreatment; FSDUP: flat sweep-frequency dispersive ultrasound pretreatment; DEUP: dual-frequency energy-gathered ultrasound pretreatment.

**Table 3 foods-11-02593-t003:** Changes in color properties of strawberry clear juice during storage after thermal pretreatment and multi-mode thermosonication.

Treatment	Time (Days)	4 °C				25 °C				37 °C			
		*L**	*a**	*b**	∆*E*	*L**	*a**	*b**	∆*E*	*L**	*a**	*b**	∆*E*
Control	0	31.84 ± 0.31 ^a^	21.24 ± 0.41 ^a^	9.32 ± 0.16 ^fg^	/	31.84 ± 0.31 ^de^	21.24 ± 0.41 ^a^	9.32 ± 0.16 ^h^	/	31.84 ± 0.31 ^d^	21.24 ± 0.41 ^a^	9.32 ± 0.16 ^ij^	/
	7	30.63 ± 0.56 ^d^	19.30 ± 0.99 ^bc^	12.27 ± 0.72 ^bc^	3.95 ± 0.31 ^ij^	32.98 ± 0.24 ^d^	18.23 ± 0.48 ^c^	14.00 ± 0.44 ^ab^	5.72 ± 0.11 ^k^	43.51 ± 0.44 ^c^	9.62 ± 0.93 ^ef^	15.02 ± 0.50 ^c^	17.53 ± 0.22 ^e^
	14	28.84 ± 0.17 ^efg^	18.50 ± 0.18 ^cde^	11.46 ± 0.22 ^cde^	4.60 ± 0.15 ^fgh^	41.80 ± 0.35 ^c^	5.15 ± 0.04 ^k^	14.21 ± 0.77 ^a^	19.56 ± 0.34 ^c^	45.17 ± 0.27 ^b^	4.23 ± 0.06 ^j^	16.21 ± 0.19 ^b^	22.69 ± 0.18 ^c^
	21	28.23 ± 0.10 ^gh^	17.60 ± 0.29 ^ef^	12.17 ± 0.16 ^bcd^	5.87 ± 0.19 ^e^	45.02 ± 0.09 ^b^	3.48 ± 0.03 ^l^	15.03 ± 0.07 ^a^	22.84 ± 0.03 ^b^	45.90 ± 0.65 ^ab^	2.75 ± 0.03 ^lm^	17.15 ± 0.38 ^a^	24.51 ± 0.50 ^b^
	28	28.11 ± 0.20 ^hi^	17.45 ± 0.54 ^efg^	13.07 ± 0.37 ^ab^	6.53 ± 0.37 ^d^	46.04 ± 0.75 ^a^	2.69 ± 0.02 ^m^	15.06 ± 0.48 ^a^	24.07 ± 0.56 ^a^	46.34 ± 0.66 ^a^	2.28 ± 0.05 ^l^	17.65 ± 0.07 ^a^	25.29 ± 0.37 ^a^
TP	0	28.47 ± 0.08 ^fgh^	18.71 ± 0.15 ^cd^	11.37 ± 0.13 ^cde^	4.69 ± 0.13 ^fg^	28.47 ± 0.08 ^gh^	18.71 ± 0.15 ^c^	11.37 ± 0.13 ^fg^	4.69 ± 0.13 ^m^	28.47 ± 0.08 ^h^	18.71 ± 0.15 ^c^	11.37 ± 0.13 ^g^	4.69 ± 0.13 ^l^
	7	27.60 ± 0.73 ^i^	15.32 ± 0.48 ^h^	8.16 ± 0.43 ^h^	7.44 ± 0.10 ^c^	27.32 ± 0.03 ^i^	14.12 ± 0.25 ^e^	8.41 ± 0.23 ^hi^	8.48 ± 0.22 ^j^	27.39 ± 0.57 ^i^	10.84 ± 0.38 ^e^	9.05 ± 0.42 ^j^	11.34 ± 0.18 ^i^
	14	26.81 ± 0.09 ^j^	13.67 ± 0.30 ^i^	8.38 ± 0.41 ^gh^	9.15 ± 0.28 ^b^	26.35 ± 0.24 ^j^	7.89 ± 0.19 ^i^	8.98 ± 0.19 ^h^	14.45 ± 0.20 ^ef^	25.38 ± 0.34 ^j^	6.92 ± 0.16 ^h^	9.65 ± 0.32 ^i^	15.72 ± 0.16 ^f^
	21	26.02 ± 0.27 ^j^	12.99 ± 0.59 ^ij^	9.05 ± 0.74 ^fgh^	9.61 ± 0.36 ^b^	25.67 ± 0.04 ^k^	6.71 ± 0.08 ^j^	9.42 ± 0.13 ^h^	15.79 ± 0.06 ^e^	24.02 ± 0.21 ^k^	5.12 ± 0.09 ^i^	9.79 ± 0.78 ^hi^	17.94 ± 0.03 ^e^
	28	25.87 ± 0.15 ^k^	12.37 ± 0.12 ^j^	9.48 ± 0.94 ^f^	10.74 ± 0.18 ^a^	24.85 ± 0.23 ^l^	4.51 ± 0.07 ^k^	9.82 ± 0.47 ^gh^	18.15 ± 0.14 ^d^	23.61 ± 0.38 ^l^	3.56 ± 0.18 ^k^	10.50 ± 0.23 ^h^	19.54 ± 0.09 ^d^
FSDUP	0	31.59 ± 0.32 ^ab^	20.32 ± 0.89 ^a^	12.84 ± 0.62 ^ab^	3.79 ± 0.47 ^ij^	31.59 ± 0.32 ^e^	20.32 ± 0.89 ^b^	12.84 ± 0.62 ^bcd^	3.79 ± 0.47 ^m^	31.59 ± 0.32 ^d^	20.32 ± 0.89 ^b^	12.84 ± 0.62 ^cde^	3.79 ± 0.47 ^m^
	7	31.23 ± 0.13 ^bc^	20.26 ± 0.18 ^ab^	10.52 ± 0.36 ^e^	1.71 ± 0.35 ^l^	30.68 ± 0.07 ^f^	17.83 ± 0.50 ^cd^	11.26 ± 0.23 ^fg^	4.11 ± 0.47 ^lm^	30.08 ± 0.19 ^ef^	13.97 ± 0.22 ^d^	11.73 ± 0.36 ^fg^	7.87 ± 0.27 ^k^
	14	29.34 ± 0.09 ^e^	18.47 ± 0.74 ^cde^	10.71 ± 0.50 ^e^	4.06 ± 0.33 ^hij^	29.31 ± 0.84 ^g^	12.19 ± 0.96 ^f^	11.78 ± 0.15 ^ef^	9.77 ± 0.68 ^i^	29.48 ± 0.14 ^fg^	9.35 ± 0.20 ^f^	11.91 ± 0.49 ^ef^	12.40 ± 0.18 ^h^
	21	28.90 ± 0.15 ^ef^	17.94 ± 0.15 ^def^	11.27 ± 0.08 ^de^	4.83 ± 0.16 ^f^	26.53 ± 0.17 ^j^	10.70 ± 0.07 ^gh^	11.81 ± 0.21 ^ef^	12.06 ± 0.04 ^g^	28.52 ± 0.22 ^h^	8.23 ± 0.25 ^g^	12.09 ± 0.70 ^ef^	13.73 ± 0.40 ^g^
	28	28.39 ± 0.17 ^fgh^	17.34 ± 0.17 ^fg^	11.11 ± 0.36 ^e^	5.52 ± 0.11 ^e^	26.34 ± 0.02 ^j^	10.19 ± 0.16 ^h^	12.21 ± 0.46 ^cde^	12.69 ± 0.14 ^g^	27.43 ± 0.04 ^i^	8.05 ± 0.25 ^g^	12.35 ± 0.13 ^de^	14.23 ± 0.21 ^fg^
DEUP	0	30.69 ± 0.05 ^cd^	20.91 ± 0.24 ^a^	13.27 ± 0.29 ^a^	4.14 ± 0.26 ^ghi^	30.69 ± 0.05 ^f^	20.91 ± 0.24 ^ab^	13.27 ± 0.29 ^bc^	4.14 ± 0.26 ^mn^	30.69 ± 0.05 ^e^	20.91 ± 0.24 ^a^	13.27 ± 0.29 ^c^	4.14 ± 0.26 ^lm^
	7	30.25 ± 0.13 ^d^	20.23 ± 0.53 ^ab^	10.64 ± 0.24 ^e^	2.36 ± 0.29 ^k^	29.33 ± 0.21 ^g^	17.52 ± 0.48 ^d^	11.51 ± 0.31 ^ef^	5.07 ± 0.37 ^kl^	29.13 ± 0.10 ^g^	13.58 ± 0.22 ^d^	11.69 ± 0.13 ^fg^	8.46 ± 0.18 ^jk^
	14	29.11 ± 0.33 ^d^	18.49 ± 0.37 ^cde^	10.53 ± 0.39 ^e^	3.52 ± 0.09 ^j^	28.14 ± 0.24 ^gh^	11.10 ± 0.16 ^g^	12.17 ± 0.26 ^cde^	11.17 ± 0.17 ^h^	28.85 ± 0.13 ^gh^	9.34 ± 0.13 ^f^	12.41 ± 0.33 ^de^	12.65 ± 0.20 ^i^
	21	28.40 ± 0.16 ^fgh^	17.22 ± 0.13 ^fg^	10.98 ± 0.13 ^e^	5.54 ± 0.16 ^e^	27.91 ± 0.07 ^hi^	8.09 ± 0.05 ^i^	12.71 ± 0.13 ^bcd^	14.14 ± 0.03 ^f^	27.31 ± 0.16 ^i^	8.15 ± 0.21 ^g^	12.98 ± 0.17 ^cd^	13.68 ± 0.15 ^g^
	28	28.34 ± 0.09 ^fgh^	16.46 ± 0.11 ^g^	11.04 ± 0.09 ^e^	6.06 ± 0.15 ^de^	27.56 ± 0.20 ^hi^	6.45 ± 0.35 ^j^	12.50 ± 0.50 ^bcde^	15.37 ± 0.29 ^e^	27.00 ± 0.11 ^i^	7.92 ± 0.05 ^g^	12.26 ± 0.69 ^de^	14.49 ± 0.16 ^g^

Note: The different lowercase superscript letters (a–m) in the same column indicate significant differences according to Duncan’s test (*p* < 0.05). TP: thermal pretreatment; FSDUP: flat sweep-frequency dispersive ultrasound pretreatment; DEUP: dual-frequency energy-gathered ultrasound pretreatment.

**Table 4 foods-11-02593-t004:** Changes in browning and clarification of strawberry clear juice during storage after thermal pretreatment and multi-mode thermosonication.

Temperature	Time (Days)	Browning (A_420_)				Clarity (T_660_)			
		Control	TP	FSDUP	DEUP	Control	TP	FSDUP	DEUP
4 °C	0	0.17 ± 0.01 ^d^	0.26 ± 0.03 ^c^	0.22 ± 0.02 ^c^	0.23 ± 0.02 ^c^	90.38 ± 0.12 ^a^	85.35 ± 0.69 ^a^	88.80 ± 0.64 ^b^	87.73 ± 0.62 ^a^
	7	0.18 ± 0.01 ^d^	0.27 ± 0.01 ^c^	0.22 ± 0.01 ^c^	0.24 ± 0.01 ^bc^	85.90 ± 0.49 ^ab^	84.26 ± 0.87 ^a^	88.67 ± 0.29 ^b^	86.12 ± 0.30 ^b^
	14	0.23 ± 0.01 ^c^	0.31 ± 0.01 ^b^	0.24 ± 0.02 ^c^	0.25 ± 0.01 ^b^	84.88 ± 0.01 ^ab^	84.91 ± 0.73 ^a^	89.30 ± 0.02 ^a^	85.42 ± 0.68 ^b^
	21	0.25 ± 0.01 ^bc^	0.33 ± 0.01 ^b^	0.28 ± 0.01 ^bc^	0.30 ± 0.02 ^ab^	78.22 ± 0.90 ^b^	83.65 ± 0.33 ^ab^	89.95 ± 0.15 ^a^	87.75 ± 0.57 ^a^
	28	0.28 ± 0.01 ^bc^	0.35 ± 0.02 ^a^	0.31 ± 0.01 ^ab^	0.33 ± 0.01 ^a^	71.25 ± 0.82 ^c^	84.35 ± 0.33 ^a^	90.28 ± 0.02 ^a^	85.68 ± 0.57 ^b^
25 °C	0	0.17 ± 0.01 ^d^	0.26 ± 0.03 ^c^	0.22 ± 0.02 ^c^	0.23 ± 0.02 ^c^	90.38 ± 0.12 ^a^	85.35 ± 0.69 ^a^	88.80 ± 0.64 ^b^	87.73 ± 0.62 ^a^
	7	0.19 ± 0.02 ^d^	0.28 ± 0.01 ^c^	0.24 ± 0.01 ^c^	0.25 ± 0.02 b	75.09 ± 0.34 ^b^	83.83 ± 0.23 ^ab^	87.50 ± 0.50 ^c^	84.70 ± 0.08 ^bc^
	14	0.24 ± 0.02 ^c^	0.33 ± 0.03 ^a^	0.25 ± 0.03 ^bc^	0.26 ± 0.02 ^b^	68.48 ± 0.03 ^c^	80.76 ± 0.42 ^b^	87.32 ± 0.55 ^c^	84.06 ± 0.22 ^c^
	21	0.31 ± 0.01 ^b^	0.35 ± 0.02 ^a^	0.30 ± 0.03 ^ab^	0.32 ± 0.02 ^ab^	62.32 ± 0.96 ^cd^	78.27 ± 0.54 ^c^	86.52 ± 0.62 ^d^	83.17 ± 0.04 ^cd^
	28	0.36 ± 0.02 ^b^	0.37 ± 0.01 ^a^	0.32 ± 0.02 ^a^	0.34 ± 0.02 ^a^	52.21 ± 0.63 ^d^	78.40 ± 0.40 ^c^	86.56 ± 0.31 ^d^	83.25 ± 0.57 ^cd^
37 °C	0	0.17 ± 0.01 ^d^	0.26 ± 0.03 ^b^	0.22 ± 0.02 ^c^	0.23 ± 0.02 ^c^	90.38 ± 0.12 ^a^	85.35 ± 0.69 ^a^	88.80 ± 0.64 ^b^	87.73 ± 0.62 ^a^
	7	0.23 ± 0.02 ^c^	0.31 ± 0.01 ^b^	0.25 ± 0.02 ^b^	0.26 ± 0.01 ^b^	71.18 ± 085 ^c^	80.84 ± 0.30 ^b^	86.12 ± 0.47 ^d^	82.66 ± 0.40 ^cd^
	14	0.27 ± 0.02 ^bc^	0.34 ± 0.02 ^a^	0.27 ± 0.02 ^ab^	0.29 ± 0.02 ^ab^	57.26 ± 0.64 ^cd^	79.31 ± 0.35 ^c^	85.47 ± 0.27 ^d^	80.78 ± 0.04 ^d^
	21	0.42 ± 0.03 ^a^	0.36 ± 0.02 ^a^	0.32 ± 0.03 ^a^	0.34 ± 0.02 ^a^	51.76 ± 0.06 ^d^	76.23 ± 0.40 ^d^	84.69 ± 0.84 ^d^	79.22 ± 0.52 ^d^
	28	0.46 ± 0.02 ^a^	0.38 ± 0.02 ^a^	0.35 ± 0.02 ^a^	0.36 ± 0.02 ^a^	45.28 ± 0.57 ^e^	76.18 ± 0.46 ^d^	84.42 ± 0.80 ^d^	79.05 ± 0.13 ^d^

Note: The different lowercase superscript letters (a–d) in the same column indicate significant differences according to Duncan’s test (*p* < 0.05). TP: thermal pretreatment; FSDUP: flat sweep-frequency dispersive ultrasound pretreatment; DEUP: dual-frequency energy-gathered ultrasound pretreatment.

## Data Availability

Data is contained within the article.

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
