# Peer review of "Effect of Multi-Mode Thermosonication on the Microbial Inhibition and Quality Retention of Strawberry Clear Juice during Storage at Varied Temperatures"

_foods, 2022, doi:10.3390/foods11172593_

Round 1
Reviewer 1 Report
Dear Authors
Manuscript entitled „ Effect of multi-mode thermosonication on the microbial inhibition, enzyme activity, and physico-chemical properties of strawberry clear juice during storage at varied temperature” which was submitted to revision in Foods is an original research study. The article adheres to the journal's standards. The experiment design is correct, the results are properly discussed. The aim of study is clear and generally the manuscript is well written (English needs polishing). The manuscript should be improved. I put my comments and suggestions:
Major:
1. What was the aim of study?
2. No discussion in the section number 3.1, 3.2, 3.3, 3.5.1, 3.6.4.
Minor:
1. I suggest include in keywords: enzymatic browning, polyphenol oxidase, strawberry juice.
2. Please add more information about preparation of strawberry clear juice, especially about raw material pretreatment, juice extraction method, process parameters etc.
3. Lines: 42-45 and lines: 46-47 they contradict each other. Please correct.
4. Lines: 47-48 “The natural polyphenol oxidase enzymes in strawberry may cause enzymatic browning” please change to ‘The polyphenol oxidase activity from strawberry (…).
5. Line 133: no information about how the enzyme activity is expressed (standard unit, katal, specific activity or residual activity?) If was standard unit please add definition of Unit.
6.Line 183: please add information about how is the content of ascorbic acid expressed.
7. Table 1: what does it mean lg CFU/mL? please explain.
8. Table 3: First column: change treatment time to storage time.
9. Figure 4: TPC - change mg/100mL to mg GAE/100 mL.
TFC - change mg/100mL to mg RE/100 mL.
10. Figure 5: TAC- change mg/100 mL to mg CGE/100mL.
11. Figure 6: Please give full units for y axis.
Author Response
Reviewer #1:
Manuscript entitled “Effect of multi-mode thermosonication on the microbial inhibition, enzyme activity, and physico-chemical properties of strawberry clear juice during storage at varied temperature” which was submitted to revision in Foods is an original research study. The article adheres to the journal's standards. The experiment design is correct, the results are properly discussed. The aim of study is clear and generally the manuscript is well written (English needs polishing). The manuscript should be improved. I put my comments and suggestions:
Response: We really appreciate your affirmative answer to this manuscript. We will try our best to revise it.
Major:
- What was the aim of study?
Response: Thanks. The aim of this study was to explore whether multi-mode thermosonication can ensure the quality stability of strawberry juice during storage, and to provide research basis for the application of multi-mode thermosonication technology in juice. To better express the aim of this study, we revised the manuscript in the abstract.
“To explore whether multi-mode thermosonication (MTS) can ensure the quality stability of straw-berry clear juice (SCJ) during storage, the effects of microbial inhibition, enzyme activity, and physico-chemical properties of SCJ pretreated by MTS were evaluated during storage at 4, 25, and 37 ℃ in comparison with thermal pretreatment (TP) at 90 ℃ for 1 min.”
- No discussion in the section number 3.1, 3.2, 3.3, 3.5.1, 3.6.4.
Response: Thanks a lot! As suggested, we have added the relevant contents into the revised manuscript. Following sentences have been added at respective sections as the discussion for the observed results:
“Fan, et al. [28] also confirmed that carrot juice processed by thermosonication at 52 °C showed significant microbial growth stability during storage at 6 °C, and consequently extend the product shelf-life.”
“Adedokun, et al. [32] also reported that the pH values of fruit juices decreased during storage due to the spoilage microorganisms.”
“Similar finding was reported by Raji, et al. [34] for thermally processed pineapple and bitter orange mixed fruits juices stored at room temperature.”
“Buvé, et al. [36] also reported that the a* value of pasteurized strawberry juices de-creased continuously during storage.
These changes may be closely related to chemical, biochemical, enzymatic, and physi-cal changes during the ultrasonic processing [15-17].”
“MTS had the effects of dissolved oxygen and enzyme inactivation, which could effectively reduce the degradation of ascorbic acid during storage [17]. Tiwari, et al. [45] also found a higher ascorbic acid retention (78.60%) for sonicated strawberry juices at the highest acoustic energy density level (0.81 W/mL) and treatment time (10 min) during 10 days of storage at 4 °C compared to untreated samples at the same storage conditions (76.20%).”
Minor:
- I suggest include in keywords: enzymatic browning, polyphenol oxidase, strawberry juice.
Response: Thanks for the suggestion. As suggested, we have revised the manuscript.
“Strawberry juice; Enzymatic browning; Polyphenol oxidase; Active ingredients; Shelf life”
- Please add more information about preparation of strawberry clear juice, especially about raw material pretreatment, juice extraction method, process parameters etc.
Response: Thanks for the suggestion. As suggested, we have added the relevant contents into the revised manuscript.
“The fresh strawberries were washed carefully to remove any adhering dirt or im-purities. The calyx and stems were also removed. Then, they were manually cut into small cubes with a sterile knife, added water at a liquid to material ratio of 2:1 mL/g, followed by juicing using a beater (HB500A, Hauswirt, China). Then, the enzymatic hydrolysis of strawberry juice was performed based on the pre-experimentally optimized enzy-matic process (Enzyme addition of 0.17%; pectinase to cellulase ratio of 3:1; enzymatic temperature of 41 oC; and enzymatic time of 35 min). After enzymatic hydrolysis and centrifuged at 8000 g for 10 min, the supernatant was filtered through two layers of gauze to obtain SCJ.”
- Lines: 42-45 and lines: 46-47 they contradict each other. Please correct.
Response: Thanks. The non-alcoholic beverages include many drinks such as carbonated drinks, juices, energy drinks, bottled water, coffee, tea and probiotic drinks. Constituents of the carbonated drinks are water, carbon dioxide and flavor. Previous studies have shown a consistent association between sugary carbonated beverage consumption and hyperuricemia. However, fruit juices are obtained by mechanical extraction (squeezing) of different fruits. The fruit juices contain many nutrients such as minerals, vitamins (especially vitamin C), antioxidants, carotenoids, phytochemicals and dietary fiber, which are essential for human health. To better express our view, we have revised the manuscript.
“The fruit juices are obtained by mechanical extraction (squeezing) of different fruits. Especially, strawberry juice contains many nutrients such as minerals, vitamins (especially vitamin C), antioxidants, carotenoids, phytochemicals and dietary fiber, which are essential for human health [5].”
- Lines: 47-48 “The natural polyphenol oxidase enzymes in strawberry may cause enzymatic browning” please change to ‘The polyphenol oxidase activity from strawberry (…).
Response: Thank you very much for your attentiveness and patience. As suggested, we have revised the manuscript.
- Line 133: no information about how the enzyme activity is expressed (standard unit, katal, specific activity or residual activity?) If was standard unit please add definition of Unit.
Response: Thanks a lot! As suggested, we have revised the manuscript.
“The results were expressed as residual enzyme activity. And the residual activity of PPO was calculated as:
(1)
where At represents the remaining enzymatic activity at time t, and A0 is the initial PPO activity of strawberry juice before treatment.”
- Line 183: please add information about how is the content of ascorbic acid expressed.
Response: Thanks a lot! As suggested, we have added the relevant contents into the revised manuscript.
“The results were expressed as mg ascorbic acid per 100 mL of juice.”
- Table 1: what does it mean lg CFU/mL? please explain.
Response: Thanks. It means that the results expressed as the logarithms of the average number of colony forming units per mL (log CFU/mL). As suggested, we have revised the manuscript.
“The results were expressed as the logarithms of the average number of colony forming units per mL (log CFU/mL).”
- Table 3: First column: change treatment time to storage time.
Response: Thanks. The first column is the treatments, and the second column is the storage time. We have adjusted the column width to distinguish.
- Figure 4: TPC - change mg/100mL to mg GAE/100 mL. TFC - change mg/100mL to mg RE/100 mL.
Response: Thanks a lot! As suggested, we have revised the manuscript.
- Figure 5: TAC- change mg/100 mL to mg CGE/100mL.
Response: Thanks for the suggestion. As suggested, we have revised the manuscript.
- Figure 6: Please give full units for y axis.
Response: Thanks for the suggestion. As suggested, we have revised the manuscript.
Reviewer 2 Report
The manuscript presents valuable and comprehensive results about the possibility of extending strawberry juice through multi-mode thermosonication. The paper is well written and complex. Some observations are made below.
The title is too long. I recommend shortening it.
Methods: Please mention the number of replicates for each determination.
Figure 2: Please number each graphic. In the second graphic of the section A Control is at 4 degree. Should it be at 25?
Table 3: Statistics are missing. Please add the letters.
When talking about correlations, please give p and r values in brackets.
Conclusion: Please add limitations and further perspective at the end.
Author Response
Reviewer #2:
The manuscript presents valuable and comprehensive results about the possibility of extending strawberry juice through multi-mode thermosonication. The paper is well written and complex. Some observations are made below.
Response: Thanks very much for your affirmative answer to this manuscript. As suggested, we have made some changes in the manuscript, and all changes in the revised manuscript were marked in blue.
The title is too long. I recommend shortening it.
Response: Thanks for the suggestion. As suggested, we have revised it as follows:
“Effect of multi-mode thermosonication on the microbial inhibition and quality retention of strawberry clear juice during storage at varied temperature”
Methods: Please mention the number of replicates for each determination.
Response: Thanks a lot! As suggested, we have added the relevant contents into the revised manuscript.
“All measurements were carried out in triplicate and the results were expressed as mean ± standard deviation.”
Figure 2: Please number each graphic. In the second graphic of the section A Control is at 4 degree. Should it be at 25?
Response: Thank you very much for your attentiveness and patience. As suggested, we have revised the manuscript.
Table 3: Statistics are missing. Please add the letters.
Response: Thanks a lot! As suggested, we have revised the manuscript.
When talking about correlations, please give p and r values in brackets.
Response: Thanks for the suggestion. As suggested, we have revised the manuscript.
Conclusion: Please add limitations and further perspective at the end.
Response: Thanks for the suggestion. As suggested, we have added the relevant contents into the revised manuscript.
“However, the changes of volatile components during storage need to be further studied to determine the accurate shelf life of SCJ.”
Reviewer 3 Report
The manuscript entitled Effect of multi-mode thermosonication on the microbial inhibition, enzyme activity, and physico-chemical properties of strawberry clear juice during storage at varied temperature, presents information related to the application of thermosonication to strawberry beverages and the evaluation of the microbial content and physico-chemical properties. The topic of the manuscript is very sound. However, the manuscript has some issues that authors must attend to prior to acceptance.
Authors must revise the language of the manuscript. There are grammar and style mistakes.
Line 4. Change temperature by temperatures.
Line 106. What is the enzymatic time? Is it reaction time? Clarify the statement.
Line 109. Frequency of 20 + 40? Were used both frequencies?
Figure 1. It is not clear the workflow of FSDUP and DEUP devices. Explain.
Section 2.7.4. Provide the detailed HPLC elution method for the quantification of ascorbic acid. Did the authors use a standard solution? Provide the information.
Line 212. Change was by were.
Table 2. How was estimated the PPO activity? In methodology, there are no equations. Which was the control treatment that represented 100 % of the enzymatic activity? What is the definition of the PPO enzymatic activity?
Sections 3.6.2 and 3.6.3. What was the rationale for analyzing spectrophotometrically the changes in total flavonoids and anthocyanins content? It is recommended to analyze the content of phytochemicals by HPLC coupled to the mass spectrometer.
Author Response
Reviewer #3:
The manuscript entitled Effect of multi-mode thermosonication on the microbial inhibition, enzyme activity, and physico-chemical properties of strawberry clear juice during storage at varied temperature, presents information related to the application of thermosonication to strawberry beverages and the evaluation of the microbial content and physico-chemical properties. The topic of the manuscript is very sound. However, the manuscript has some issues that authors must attend to prior to acceptance.
Response: Thanks very much for your suggestions and comments for enhancing our work. We will try our best to revise the manuscript.
Authors must revise the language of the manuscript. There are grammar and style mistakes.
Response: Thanks a lot! As suggested, we have revised the manuscript.
Line 4. Change temperature by temperatures.
Response: Thank you very much for your attentiveness and patience. As suggested, we have revised the manuscript.
Line 106. What is the enzymatic time? Is it reaction time? Clarify the statement.
Response: Thanks a lot! The enzymatic time is enzymatic reaction time. As suggested, we have revised the manuscript.
Line 109. Frequency of 20 + 40? Were used both frequencies?
Response: Yes. 20+40 kHz means that the ultrasound transducer of two frequencies works simultaneously. To better express the ultrasound mode, we have explained it by this sentence: “flat sweep-frequency dispersive ultrasound pretreatment (FSDUP) was done at 60 oC for 15 min under the frequency of 20+40 kHz in simultaneous operation mode”
Figure 1. It is not clear the workflow of FSDUP and DEUP devices. Explain.
Response: Thanks a lot! As suggested, we have revised the manuscript in Fig. 1.
Section 2.7.4. Provide the detailed HPLC elution method for the quantification of ascorbic acid. Did the authors use a standard solution? Provide the information.
Response: Thanks for the suggestion. An isocratic elution procedure was used in this study. Identification was performed by comparison with the retention time of ascorbic acid standards and quantification of ascorbic acid in strawberry juice by external calibration methods. As suggested, we have added the relevant contents into the revised manuscript.
“The HPLC operational parameters were as follows: column temperature of 25 oC, the mobile phase of 0.1% oxalic acid isocratic solution, the flow rate of 0.8 mL/min, the detection wavelength of 254 nm and the injection volume of 20 μL. Ascorbic acid of SCJ was identified by comparisons of the retention time (tR=7.799 min) with ascorbic acid standard and the concentration was quantified by the external calibration method. The calibration curve of ascorbic acid (y=85838x-60180, R2=0.9998) was constructed by plotting the peak areas versus the concentrations (from 0 to 0.25 g/L) of the standard compound.”
Line 212. Change was by were.
Response: Thanks a lot! As suggested, we have revised the manuscript.
Table 2. How was estimated the PPO activity? In methodology, there are no equations. Which was the control treatment that represented 100 % of the enzymatic activity? What is the definition of the PPO enzymatic activity?
Response: Thanks for the suggestion. As suggested, we have added the relevant contents into the revised manuscript.
“The unit of enzyme activity was defined as the amount of enzyme that causes a change of 0.001 absorbance units per minute, and PPO activity was expressed as ∆OD/(mL·min). The results were expressed as residual enzyme activity. And the residual activity of PPO was calculated as:
(1)
where At represents the remaining enzymatic activity at time t, and A0 is the initial PPO activity of strawberry juice before treatment.”
Sections 3.6.2 and 3.6.3. What was the rationale for analyzing spectrophotometrically the changes in total flavonoids and anthocyanins content? It is recommended to analyze the content of phytochemicals by HPLC coupled to the mass spectrometer.
Response: Thanks a lot! The 3-hydroxy, 4-hydroxy or 5-hydroxy, 4-carbonyl or o-bisphenol hydroxyl in flavonoids reacted with aluminum salts to form red complexes under alkaline conditions. The absorbance was measured at 510 nm by spectrophotometry. Compared with rutin standard, the content of total flavonoids in strawberry juice was determined. Anthocyanins change color due to structural changes under different pH conditions. They appear red under acidic conditions and blue under alkaline conditions. The structural transformation of anthocyanin chromophores is a function of pH, while the characteristic spectra of interfering substances do not change with pH. The main anthocyanin in strawberry juice is cyanidin-3-glucoside, which forms red 2-benzopyranoid compounds at pH 1.0 and has the maximum absorption at 510 nm. At pH 4.5, it exists in the form of colorless methanol pseudobase. Therefore, the differential pH method can be used to determine the anthocyanin content in strawberry juice as represented by coryphyllin-3-glucoside. Spectrophotometry has the advantages of high sensitivity, simple operation and rapidness. As suggested, we will use HPLC coupled to the mass spectrometer to analyze the content of phytochemicals in future research.
Round 2
Reviewer 3 Report
The manuscript has been improved. I recommend the acceptance of the manuscript.